# Physiologically Based Pharmacokinetic (PBPK) Modeling for Predicting Brain Levels of Drug in Rat

**DOI:** 10.3390/pharmaceutics13091402

**Published:** 2021-09-03

**Authors:** Bárbara Sánchez-Dengra, Isabel Gonzalez-Alvarez, Marival Bermejo, Marta Gonzalez-Alvarez

**Affiliations:** Engineering: Pharmacokinetics and Pharmaceutical Technology Area, Miguel Hernandez University, San Juan de Alicante, 03550 Alicante, Spain; barbarasanchezdengra@gmail.com (B.S.-D.); isabel.gonzalez@umh.es (I.G.-A.); marta.gonzalez@umh.es (M.G.-A.)

**Keywords:** blood−brain barrier (BBB), physiologically based pharmacokinetics (PBPK), quantitative structure–property relationships (QSPRs), distribution volume in brain (V_u,brain_), plasma−brain partition coefficient (Kp_uu,brain_)

## Abstract

One of the main obstacles in neurological disease treatment is the presence of the blood–brain barrier. New predictive high-throughput screening tools are essential to avoid costly failures in the advanced phases of development and to contribute to the 3 Rs policy. The objective of this work was to jointly develop a new in vitro system coupled with a physiological-based pharmacokinetic (PBPK) model able to predict brain concentration levels of different drugs in rats. Data from in vitro tests with three different cells lines (MDCK, MDCK-MDR1 and hCMEC/D3) were used together with PK parameters and three scaling factors for adjusting the model predictions to the brain and plasma profiles of six model drugs. Later, preliminary quantitative structure–property relationships (QSPRs) were constructed between the scaling factors and the lipophilicity of drugs. The predictability of the model was evaluated by internal validation. It was concluded that the PBPK model, incorporating the barrier resistance to transport, the disposition within the brain and the drug–brain binding combined with MDCK data, provided the best predictions for passive diffusion and carrier-mediated transported drugs, while in the other cell lines, active transport influence can bias predictions.

## 1. Introduction

The brain is the most important organ in living beings as it controls their vital functions: nutrition, interaction and reproduction. The importance of the brain means all the capillaries that supply it with oxygen and nutrients are composed of extremely tight endothelial cells surrounded by a thick layer of astrocytes and pericytes, all of this to prevent dangerous substances from reaching it. This group of protective and regulating cells is known as the blood–brain barrier (BBB).

Besides the physicochemical protection of the BBB, the brain is also protected by the cerebrospinal fluid (CSF), a constantly secreted liquid that helps to maintain the brain’s homeostasis for normal neurological function, acts as a cushion between the brain and the skull, and makes the central nervous system (CNS) apparently “lighter”, as it is floating in this liquid [1]. There are two different barriers that separate the CSF from the rest of the body—the blood–cerebrospinal fluid barrier (BCSFB), which is around the capillaries within brain ventricles, and the blood–arachnoid barrier, around the subarachnoid space [2,3]. 

The extra protection of the CNS hinders the development of new drugs for the treatment of neurological pathologies (glioblastoma, Alzheimer’s disease, depression, Parkinson’s disease or epilepsy, among others), as molecules experience difficulties in reaching their target. Because of this, the pharmaceutical industry has developed several strategies to bypass the BBB: chemical strategies, such as the development of prodrugs or chemical drug delivery systems (CDDS), physical strategies, such as the use of ultrasounds to temporarily open the tight junctions of the endothelial cells, or nanotechnological strategies, which include developing lipid-based, polymer-based or metal-based nanocarriers [4].

On the other hand, the use of reliable methods to evaluate whether a new candidate will reach the brain is as important as the development of new strategies to bypass the BBB. This type of method can include in silico, in vitro, in situ or in vivo tests; however, except for the in vivo tests that directly measure the brain concentration of a drug over time, the results obtained with the other methodologies have been considered controversial, as they do not sufficiently accurately reflect whether the drug will reach the brain and perform its action [5,6].

Frequently, researchers use the permeability clearance into the brain (*Cl_in_*), or the influx permeability surface area product (*PS_in_*), which is the same, to define whether or not a drug will permeate the BBB, or at least to say if a new delivery system will improve on the action of a previous one [6]. Nevertheless, this parameter is not the only one that should be used to define the potential success of a drug in the treatment of a neurological pathology, because it gives a measurement of the rate of transport through the BBB; however, fast permeation does not mean fast action or a better performance, as concentrations in the brain are also influenced by (A) the binding of the drug to the tissue once it crosses the BBB, (B) the binding of the drug to the proteins in blood, and (C) the presence of efflux transporters that remove the drug from the brain [5]. According to Dagenais et al., the *PS_in_* of loperamide obtained by the in situ brain perfusion method was 98.6 ± 17.3 μL/min × g brain, while, the value of the same parameter, obtained by the same method, for morphine was 10.4 ± 3 μL/min × g brain; thus, loperamide is more permeable than morphine through the BBB, although morphine is much more active in the CNS. This gives a clear example of the lack of reliability of the influx clearance parameter when it is taken on its own [5,7].

When the efflux clearance (*Cl_out_*) is taken into account as well as the unbound fractions of drug, then the other two neuropharmacokinetic (neuroPK) parameters can be defined too: the unbound plasma–brain partition coefficient (*Kp_uu,brain_*) and the apparent volume of distribution in the brain (*V_u,brain_*) [5,8].

The *Kp_uu,brain_* is defined as the relationship between the concentration of free drug in plasma (*C_u,p_*) and the concentration of free drug in the brain (*C_u,b_*) at a steady state. It can be obtained according to Equation 1, in which the *AUC_u_* is the area under the unbound concentration versus time curve. This parameter includes the passive and active transport of the unbound drug through the BBB in both directions (influx and efflux); because of this, and considering that only the free fraction of the drug can cross the BBB, the *Kp_uu,brain_* is judged as a more informative parameter than the *Kp_brain_* (also known as logBB), a previously described parameter that used total concentrations to evaluate the permeability through the BBB [5,9,10].
(1)$$Kp_{uu,brain}=\frac{AUC_{u,brain}}{AUC_{u,plasma}}=\frac{Cl_{in}}{Cl_{out}}=\frac{PS_{in}}{PS_{out}}$$

The *V_u,brain_*, meanwhile, reflects the distribution of the drug once it has crossed the BBB, and can be defined as a relationship between the total amount of drug present in the brain and the concentration of free drug in the brain (*C_u,b_*). If the drug has a high affinity for brain tissue, its unbound fraction of drug in the brain (*f_u,brain_*) will decrease, and therefore, the *V_u,brain_* will be greater, as shown in Equation (2), where *V_ECF_* is the volume of extracellular fluid and *V_ICF_* is the volume of intracellular fluid [11]. A higher *V_u,brain_* can be translated into a longer half-life of the drug in the brain, independently of its transport through the BBB or its plasma concentration [5]. So, it is particularly helpful to know this parameter or the *f_u,brain_* when a new candidate for the treatment of a neurological disease is evaluated.
(2)$$V_{u,brain}=V_{ECF}+\frac{1}{f_{u,brain}}\cdot V_{ICF}=0.2+\frac{1}{f_{u,brain}}\cdot 0.6\;\;\;\;\;\left(mL/g\;brain\right)$$

Besides the different tests for obtaining the parameters mentioned above, over the last few years, researchers have also worked on the development of mathematical models, such as physiologically based pharmacokinetic (PBPK) models, for describing and predicting the behaviour of CNS drugs after being administered. In 2019, Vendel et al. published a review wherein they summarized different types of mathematical models for describing drug distribution within the brain, and all the processes and properties that should be taken into account to develop the “perfect” model [2]:Brain-specific properties, such as the properties of the brain vascular network and the different brain barriers, the characteristics of the brain tissue and the CSF, the fluid movements within the brain or the presence of metabolic enzymes in the CNS;Drug-specific properties, such as molecular (molecular weight, polar surface area, shape or number of hydrogen bond donors and acceptors), physicochemical (pKa, solubility or lipophilicity) and pharmacokinetic properties;Processes affecting drug distribution within the brain, e.g., the drug transport through the brain vascular system, the brain barriers or within the brain fluids, the drug extra-/intracellular exchange, the drug binding or the drug metabolism.

It is shown that, despite the high number of models available, some are too complex, others are not completely predictable, and none are able to cover all the processes mentioned above. At the end of the review, it is concluded that there are three important points that a neuro-PBPK model should include in order to give accurate predictions of the in vivo behaviour: (A) the barrier transport, (B) the transport of the drug once in the brain, and (C) drug–brain binding [2].

Normally, neuroPK parameters (*PS_in_*, *Kp_uu,brain_* and *V_u,brain_*) are obtained using different experimental models. In this work, a previously proposed and validated single in vitro system [11,12] was used to derive those parameters. Later, these in vitro data were combined with in silico and in vivo pharmacokinetic information from outside the brain to develop a new neuro-semi-physiological mathematical model, including the three factors mentioned above (the barrier transport, the transport within the brain and the drug–brain binding). Thus, the main objective of this work was to jointly develop a new in vitro system coupled with a physiological-based pharmacokinetic (PBPK) model able to predict brain concentration levels for different drugs in a rat. Furthermore, the predictability of the new PBPK model was compared across different cell lines chosen for obtaining the in vitro neuroPK parameters.

## 2. Materials and Methods

### 2.1. Drugs and Products

The six drugs used for constructing the mathematical model (amitriptyline, caffeine, carbamazepine, fleroxacin, pefloxacin and zolpidem) and HPLC-grade solvents (acetonitrile, methanol and water) were purchased from Sigma-Aldrich (Barcelona, Spain). The MDCK cell line was purchased from ATCC (USA), MDCK-MDR1 cells were provided by Dr. Gottessman, and the MM (National Institutes of Health, Bethesda, MD, USA) and hCMEC/D3 cell lines were purchased from Cedarlane (Burlington, ON, Canada). Pig brain homogenate was kindly supplied by a local slaughterhouse.

Dulbecco’s modified eagle’s medium (DMEM) with a high content of glucose, L-glutamine, HEPES, MEM non-Essential aminoacid, penicillin−streptomycin, trypsin-EDTA, Hank’s balanced salt solution (HBSS) and fetal bovine serum (FBS) for the cell culture of MDCK and MDCK-MDR1 cell lines were purchased from Sigma-Aldrich (Barcelona, Spain).

The products needed for the culture of hCMEC/D3 cells were purchased from Sigma-Aldrich (Barcelona, Spain) (hydrocortisone, ascorbic acid, HEPES, Triton X-100 and bFGF), Gibco (Barcelona, Spain) (FBS, penicillin–streptomycin, chemically defined lipid concentrate, HBSS, collagen I rat protein and trypsin-EDTA), Lona (Barcelona, Spain) (EBM-2 medium).

### 2.2. Cell Culture and Permeability Studies

MDCK and MDCK-MDR1 cells were cultured and seeded according to the protocol explained in [11], while the culture and seeding of hCMEC/D3 cell line was done as explained in [12].

On the day of the experiment, the cells were seeded in 6-transwell plates (effective area: 4.2 cm^2^, pore size: 0.4 micron and pore density: 100 ± 10 × 10^6^/cm^2^), which were supposed to be confluent, washed twice with HBSS, and transepithelial electrical resistance (TEER) was measured. If the TEER values were around 30–40 kΩ·cm^2^ [13] for hCMEC/D3 cells, around 130–150 kΩ·cm^2^ [13] for MDCK cells and around 120–140 kΩ·cm^2^ [13] for MDCK-MDR1 cells, then the integrity of the monolayers was considered appropriate, and three types of experiments were carried out according to [11,12]:Standard AB. This experiment was designed for obtaining the apparent influx permeability (*P_app A→B_*). The drug was dissolved in HBSS; this solution was put into the apical chamber of the transwell and the basolateral chamber was filled with cleaned HBSS. Four samples were taken from the basolateral chamber at pre-established times (15, 30, 60 and 90 min) [11,12]. Three replicates were carried out for each drug;Standard BA. In this case, the montage was the opposite to the first condition and the basolateral chamber was filled with a drug solution in HBSS, while the apical chamber was filled with cleaned HBSS. Four samples were taken from the apical chamber at pre-established times (15, 30, 60 and 90 min) [11,12]. With this test, the apparent efflux permeability (*P_app B→A_*) was obtained. Three replicates were carried out for each drug;Brain homogenate BA. This last condition gives the free drug apparent efflux permeability (*P_app HOM_*), as the drug is added to the basolateral chamber after being dissolved in a 1:3 pig brain homogenate:phosphate buffer (180 mM, pH 7.4) solution, and only the free fraction of drug will be able to cross to the apical chamber, where 4 samples were taken at pre-established times (15, 30, 60 and 90 min) [11,12]. Three replicates were carried out for each drug.

The neuroPK parameters (*Kp_uu,brain_*, *f_u,brain_* and *V_u,brain_*) were obtained by means of the combination of the apparent permeabilities mentioned above with Equations (1), (3) and (2), respectively.
(3)$$f_{u,brain}=\frac{P_{app\;B\to A}}{P_{app\;HOM}}$$

### 2.3. HPLC Analysis of the Samples

Samples from brain homogenate experiments were diluted (50:50) with cold methanol to precipitate proteins, and all the samples were centrifuged at 10,000 rpm for 10 min. Then, the supernatants were analyzed by HPLC. An UV-HPLC equipment (Barcelona, Spain) (Waters 2695 separation module and Waters 2487 UV detector) with a column XBridge C18 (3.5 μM, 4.6 × 100 mm) (Barcelona, Spain), a flow rate of 1 mL/min, a run temperature of 30 °C and an injection volume of 90 μL was used for the analysis. Appendix A summarizes the rest of the chromatographic conditions. All analytical methods were validated and demonstrated to be adequate regarding linearity, accuracy, precision, selectivity and specificity.

### 2.4. Model Construction

Figure 1 shows the scheme of the semi-physiological model. An extra absorption site was added when needed (if the available in vivo data were not from intravenous administration).

We defined one compartment for plasma and two compartments for the CNS (one representing the brain tissue itself and the other representing the CSF). According to the scheme, drugs can go in or out of the CNS through the BBB or through the BCSFB, depending on the area that they reach or leave (the tissue or the CSF). Additionally, it was considered that drugs can pass from brain tissue to the CSF following the bulk flow (*Q_bulk_*) of the ECF to the CSF, and they can also return to plasma with the drainage of the CSF (*Q_sink_*). As CSF is considered a clear liquid, all the drug concentration inside it was considered the unbound drug concentration, while in the plasma and brain tissue, the unbound concentrations were considered equal to the unbound fraction of the drug in that compartment multiplied by the total concentration (the *f_u,brain_* that was used in this calculation was obtained from the permeability studies). Influx and efflux clearances through the different brain barriers were obtained by multiplying the apparent permeabilities of the in vitro studies by the surface areas of those barriers. Differential Equations (4)–(6) describe the processes mentioned above for the administration of an intravenous single dose. The values of the physiological parameters, brain volume (*V_b_*), CSF volume (*V_CSF_*), *Q_bulk_*, *Q_sink_*, BBB surface area (*S_BBB_*) and BCSFB surface area (*S_BCSFB_*) were fixed and equal for all drugs, according to Ball et al. (*V_b_* = 1.28 cm^3^, *V_CSF_* = 0.25 cm^3^, *Q_bulk_* = 0.012 cm^3^/s, *Q_sink_* = 0.132 cm^3^/s, *S_BBB_* = 187.5 cm^2^) and Engelhard et al. (*S_BCSFB_* = 0.0375 cm^2^) [14,15].
(4)$$V_{d}\cdot \frac{dC_{p}}{dt}=-PS_{BBBin}\cdot C_{u,p}+PS_{BBBout}\cdot C_{u,b}-PS_{BCSFBin}\cdot C_{u,p}+PS_{BCSFBout}\;\cdot C_{CSF}+Q_{sink}\cdot C_{CSF}-k_{el}\cdot C_{p}\cdot V_{d}$$
(5)$$V_{b}\cdot \frac{dC_{b}}{dt}=PS_{BBBin}\cdot C_{u,p}-PS_{BBBout}\cdot C_{u,b}-Q_{bulk}\cdot C_{u,b}$$
(6)$$V_{CSF}\cdot \frac{dC_{CSF}}{dt}=PS_{BCSFBin}\cdot C_{u,p}-PS_{BCSFBout}\cdot C_{CSF}-Q_{sink}\cdot C_{CSF}+Q_{bulk}\cdot C_{u,b}$$

Plasma and brain concentration profiles were obtained from the literature [16,17,18,19,20] for the six drugs studied. Depending on the publication, the brain concentration profiles derived were for free drug in the brain, total drug in the brain or drug in the ECF (which was considered equivalent to free drug in the brain). The type of data found does not affect the objective of this study, as both free and total drug concentrations are defined in the model.

When the drug administration was performed extravascularly, Equation (7) was added to the model, and Equation (4) was substituted by Equation (8), in which *k_a_* is the absorption rate constant. On the other hand, if an intravenous infusion was the method of administration, Equation (4) was substituted by Equation (9), where *k_0_* is the infusion rate constant.
(7)$$\frac{dA}{dt}=-k_{a}\cdot A$$
(8)$$V_{d}\cdot \frac{dC_{p}}{dt}=k_{a}\cdot A-PS_{BBBin}\cdot C_{u,p}+PS_{BBBout}\cdot C_{u,b}-PS_{BCSFBin}\cdot C_{u,p}\;+PS_{BCSFBout}\cdot C_{CSF}+Q_{sink}\cdot C_{CSF}-k_{el}\cdot C_{p}\cdot V_{d}$$
(9)$$V_{d}\cdot \frac{dC_{p}}{dt}=k_{0}-PS_{BBBin}\cdot C_{u,p}+PS_{BBBout}\cdot C_{u,b}-PS_{BCSFBin}\cdot C_{u,p}+PS_{BCSFBout}\;\cdot C_{CSF}+Q_{sink}\cdot C_{CSF}-k_{el}\cdot C_{p}\cdot V_{d}$$

Then, the plasma and brain profiles were adjusted to the model with the Berkeley-Madonna^®^ software (Berkeley, CA, USA), assuming that three scaling factors were needed for transforming the in vitro data obtained from cell cultures into in vivo information (Equations (10)–(14)).
(10)$$PS_{BBBin}=SC_{1}\cdot P_{app\;A\to B}\cdot S_{BBB}$$
(11)$$PS_{BCSFBin}=SC_{1}\cdot P_{app\;A\to B}\cdot S_{BCSFB}$$
(12)$$PS_{BBBout}=SC_{2}\cdot P_{app\;B\to A}\cdot S_{BBB}$$
(13)$$PS_{BCSFBout}=SC_{2}\cdot P_{app\;B\to A}\cdot S_{BCSFB}$$
(14)$$C_{u,b}=SC_{3}\cdot f_{u,brain}\cdot C_{b}$$

Most mathematical predictive models use scaling factors to transform the in vitro data/parameters into in vivo data due to the different characteristics between the in vitro cells/tissues used and the whole organism; for example, physiological characteristics, such as the surface area available for transport. Therefore, every in vitro parameter used in the model needs to have a scaling factor [21,22]. The first and second scaling factors (SC1 and SC2) were used to adjust the in vitro apparent permeabilities to the physiological permeability, as in [14], because of the differences in terms of surface area and transporters between the in vitro tests and rat physiology. Moreover, the third scaling factor (SC3) was used to correct the possible deviation present in the in vitro *f_u,brain_* values, due to the different composition and behavior of the pig brain homogenate and a healthy rat brain. The use of scaling factors allows researchers to obtain an empiric approximation of how a drug accesses the brain. Later on, a refined mechanistic model may be obtained by including both passive and active transport parameters in the differential equations and the expression level differences between the in vitro and in vivo situation.

Table 1 shows the values of the rest of the PK parameters, which were used in the model and were different for each drug. Due to the difficulties of obtaining research articles with all the PK information for rats, in some cases, parameters had to be calculated from human data. The *V_d_* of drugs marked with a + symbol (Table 1) was calculated using the human value (L/kg), and then multiplying it by the weight of the rats used in the studies (from which the plasma and brain profiles were obtained). The rats’ weights varied between 250 and 300 g.

### 2.5. Quantitative Structure–Property Relationships (QSPRs)

Once all the profiles were adjusted and the three scaling factors were defined for each drug, they were related with their lipophilicity in order to obtain three different QSPRs for each cell line. We are aware that the scaling factors could be influenced by several physicochemical properties, such as lipophilicity, molecular weight, polar surface area, *pKa*, etc.; nevertheless, the reduced number of compounds precluded the evaluation of complex models. As lipophilicity seems to be the main factor affecting membrane permeation, we selected this parameter to explore its influence. Appendix A shows the molecular and physicochemical properties of the six drugs studied, as well as the transporters for which they are substrates.

Finally, an internal validation of the model was performed to evaluate its predictability in the different cell lines. So, the adjusted scaling factors were substituted in the model by the ones predicted from the QSPRs, and the prediction error percentages (PE%) for brain *C_max_* and brain AUC were calculated according to Equation (15).
(15)$$PE\%=\left|\frac{Experimental\;value-Predicted\;value}{Experimental\;value}\right|\cdot 100$$

## 3. Results

Apparent permeability values and the *f_u,brain_* obtained from the in vitro tests with MDCK, MDCK-MDR1 and hCMEC cell lines are summarized in Table 2. These values were used later as fixed parameters in the model for obtaining the simulated plasma and brain profiles (Figure 2).

As seen in Figure 2, the model is able to describe the behavior of the six model drugs, in terms of both plasma and brain concentration profiles, after adjusting the initial *V_d_*, *k_a_*, *k_el_* and the scaling factors (SC1, SC2 and SC3), which were defined initially as one. Table 3 shows the values of the parameters mentioned above after being fitted for the different drugs and cell lines.

With the aim of seeing how accurate the adjustment was, the initial PE% values were calculated. The first row of Table 4 shows the mean PE% for brain *C_max_* and brain the AUC for each cell line; due to the similarity of all the values, it can be seen that the model adjusts properly to all the cell lines.

Figure 3 shows the three QSPRs developed for each cell line after combining the logP of each drug with the natural logarithm of each scaling factor (LnSC).

When the internal validation of the model was complete, the equations of each QSPR were used to calculate the predicted LnSC, which were transformed into SC and substituted into the model to obtain the simulated profiles. Figure 4 shows the simulated brain profiles and visual predictive checks for each drug after using the QSPR equations, and in the second row of Table 4, the PE% values for the *C_max_* and the AUC for these brain predictions are summarized.

Although all the PE% values were increased from the fitted profiles to the simulated ones, all of them remained below 50%, which can be considered appropriate due to the complexity of the model, which mixes in silico, in vitro and in vivo data for obtaining brain profiles. Furthermore, a clear tendency can be seen when both the PE% values of each cell line are added: as the complexity of the cell line increases, the error increases, which could be explained by the influence that transporters have on the SC of the drugs that are substrates of them, and not on the passive drugs.

## 4. Discussion

The development of new in vitro and mathematical tools able to predict concentration levels of drugs in humans is extremely interesting for pharmaceutical industries, as they can help avoid failures after huge investments of money, and they can also bypass ethical problems. With this aim, over many years, several researchers have worked on the development of new in vitro dissolution tools or in vitro–in vivo correlations in the field of oral absorption, and biowaivers have been highly legislated [30,31,32,33]. In terms of CNS medicine, although it is less regulated, the development of new PBPK models has become even more important, due to the difficulties in obtaining new treatments able to cross the BBB. The major problem, however, in obtaining this type of neuroPK tool is the lack of human in vivo data to compare with, as obtaining brain profiles is not as simple as obtaining plasma ones. So, in this work, a new PBPK model was developed as an initial tool to predict the brain levels of drugs in rats.

Table 2 shows the values of the apparent permeabilities (*P_app A→B_* and *P_app B→A_*) and the *f_u,brain_* obtained with the MDCK, the MDCK-MDR1 and the hCMEC/D3 cell lines. *P_app A→B_* corresponds to the Cl_in_ of the drug through the BBB. With the combination of *P_app A→B_* and *P_app B→A_*, the *Kp_uu,brain_* can be obtained, and using Equation (2), the in vitro *f_u,brain_* parameter can be translated to *V_u,brain_*. As such, the three most important neuroPK parameters (*Cl_in_*, *Kp_uu,brain_* and *V_u,brain_*) used to describe the rate and extent of drug delivery to the brain can be obtained with the same methodology, a fact that has already been highlighted in [11] and [12], investigations in which it was proven that the in vivo neuroPK parameters could be predicted from the in vitro ones. In 2013, the methodology was established in MDCK and MDCK-MDR1, cell lines and in 2021 the predictions were compared with the ones obtained in hCMEC/D3 using the same methodology; in all cases, an adequate *r^2^* was obtained [11,12]. The *P_app A→B_*, *P_app B→A_* and *f_u,brain_* values in Table 2 are variable from one cell line to the other, which could be due to the morphological and physiological differences between them, in terms of both tight junctions and transporters [34,35,36].

For obtaining an ideal in vitro BBB model, an environment reflecting the extracellular matrix and the cells surrounding the endothelial cells that constitute the BBB vessels should be simulated. Nonetheless, this environment has not yet been obtained, and cell monolayers lack of pericytes, astrocytes, neurons or different constituents of the neuroglia vascular unit (NGVU). Because of this, in vitro models can express some of the different transporters present in the BBB, but not all. Furthermore, depending on the cell type, in vitro models can express more or less tight junctions, avoiding the passage of drugs through the BBB [37]. The main advantage of cell monolayers is their utility for high-throughput screening [38].

The variability in the in vitro parameters is also reflected when the profiles are fitted with Berkeley-Madonna^®^ and the three scaling factors are obtained for each cell line, as seen in Table 3. The use of scaling factors when establishing a PBPK model of the CNS was previously seen in other studies, as in [14]. Ball et al. defined two scaling factors for transforming in vitro Caco-2 permeability values into the influx and efflux BBB rate constants [14]. In this work, three different scaling factors were defined:SC1 and SC2 were equivalent to those defined by Ball et al., as they transform the apparent permeabilities of the MDCK, MDCK-MDR1 and hCMEC/D3 cell lines into influx and efflux clearances through the BBB. The main justification for introducing these scaling factors is the difference between the rat BBB and the in vitro monolayers. For instance, in terms of tight junctions, it has been proven that primary rat brain endothelial cell cultures have high levels of occludin, endothelial cell-specific adhesion molecule (ESAM) and claudin-5, while in MDCK and MDCK-MDR1 cells, the most abundant proteins are claudin-1 and claudin-2, and in hCMEC/D3, claudin-11 [34]. Furthermore, there are differences in the morphology of the different cell lines and in transporter expressions [34];The reasons for adding SC3, which re-scale the in vitro *f_u,brain_*, were, on the one hand, to bypass the inter-species differences, as the brain homogenate used in the in vitro studies came from pigs and the model was to be used to predict rat brain profiles, and, on the other hand, to correct the possible error present in the parameter due to the homogenization process, as brain structures are broken and membrane and internal proteins get mixed.

These scaling factors have been shown to allow the model to fit to the profiles regardless of the origin of the in vitro data, as seen in Figure 2, where the MDCK, MDCK-MDR1 and hCMEC/D3 cell lines are all overlapping in both plasma and brain profiles. Nevertheless, these scaling factors are not useful if they cannot be known in advance when a new drug is developed, because the brain profiles could not be predicted with the current model. Because of this, three different QSPRs were developed for each cell line (Figure 3), relating the lipophilicity of the drugs to the natural logarithm of the scaling factors; thus, in the future, the scaling factors for a new drug could be predicted from its logP.

In Figure 3, two clear non-linear tendencies can be seen: for SC1 and SC2, as the lipophilicity of drugs increases, the value of the scaling factors increases too; for SC3, the relationship follows a parabolic function, and, firstly, the value of the scaling factor increases with logP, but then it arrives to a top zone, and then it decreases.

Sigmoidal and parabolic relationships with lipophilicity have been described for parameters such as intestinal and gastric membrane permeability [39,40,41]. In the case of the intestinal barrier, its lipidic nature favors the permeation of lipidic compounds up to the limit, due to unstirred water layer diffusion. We have not observed any asymptote in our correlations, but this could be due to the moderate lipophilicity of the assayed drugs. In the case of the gastric barrier, the combined alternate presence of hydrophilic and lipophilic layers determines a parabolic correlation between lipophilicity and gastric permeation. As SC3 is related to the differences in composition of the used brain homogenates, which contain different balances of hydrophilic/lipophilic components, a parabolic relationship is a reasonable approach.

In all cases, the tendency is clear and, except for one of the QSPRs (the one for the SC1 in the hCMEC/D3 cell line), the *r^2^* of all of them is greater than 0.80. logP was chosen as the property to relate to the scaling factors because it is one of the major descriptors used by researchers for both calculating BBB permeability [10,42] and obtaining a theoretical unbound fraction of the drug in plasma or brain [42,43].

However, to conclude on whether a given mathematical model is good enough to fulfil its design, it is not enough to look at the *r^2^* parameter of its correlations, but it is necessary to evaluate its predictability by internal validation, comparing the predictions that the model gives with the experimental data that were used for obtaining it. Figure 4 shows, therefore, the simulated brain profiles for the drug studied. In this Figure, it can be seen that there are differences in the predictions between the different cell lines; in fact, when the fitted profiles were presented (Figure 2), all the MDCK, MDCK-MDR1 and hCMEC/D3 profiles were shown to be overlapping, but now, in some cases, they are quite different. Regarding the MDCK cell line, it can be seen that the simulated profiles are, in all drugs, near to the experimental line; meanwhile, the worst prediction for the MDCK-MDR1 cells is found to be that for carbamazepine, a drug substrate of the ABCC2 and RALBP1 transporters [25], and in the case of hCMEC/D3, the zolpidem (passively transported [25]) profile is clearly overpredicted. These clear prediction errors in Figure 4, whose quantitative values (from Equation (15)) are shown in Table 4, confirm that the simulated profiles are worse than the fitted ones, and the best profiles are the ones obtained with the MDCK cell line.

The MDCK cell line is a fairly simple model of BBB, defined as a surrogate for in vivo permeability studies because of its tight junctions, but it does not have significant levels of any BBB transporter [6]. Because of this, some transfected cell lines of MDCK have been created, such as the MDCK-MDR1, which has the benefits of MDCK’s tight junctions, but incorporates the P-glycoprotein (ABCB1) efflux transporter, or the MDCK-MRP2 cell line, which incorporates the multidrug resistance-associated protein 2 transporter (ABCC2) [44]. On the other hand, hCMEC/D3 cells are much more complex, and, despite being less tight in vitro, they contain many more transporters [34,45]. As such, it is thought that they are the most reliable for obtaining brain profile predictions. However, in this study, it was seen that, as the complexity of the cell line was increased, the predictions deteriorated (Table 4). A potential explanation for this fact is that the proposed PBPK model does not include any active transporter in its differential equations, so all the active components of BBB penetration are summarized in the in vitro parameters employed (*P_app A→B_*, *P_app B→A_* and *f_u,brain_*), and thus in the three scaling factors. The model would be more physiological if the active components were included in the equations, but the passive pathway with a scaling factor was assumed as a simplification, because only one concentration of each drug was tested, and for modeling the active transport, at least three concentrations of each drug must be studied [21]. Thus, with the scaling factor, an over-parameterization of the model was avoided. Different types of drugs were used in this study (Appendix A); some of them with theoretical passive access to the BBB and others of different transporters. In consequence, when the MDCK cell line is used, as it has no transporters, all the scaling factors contain only a passive component. On the other hand, in the other two cell models, the scaling factors vary depending on their transport; the passive drugs will only have these components in their SCs, while the drugs that interact with a transporter will have both the passive and active components in their SCs. Furthermore, the active component of the SC will be different depending on the transporter, so in the hCMEC/D3 cell line, in which there are more transporters, the variability will be greater than in MDCK-MDR1, where the only significant transporter is the P-gp.

In summary, in this study, a new PBPK model, which is able to predict the brain profiles of different types of drugs (both passive and substrate of transporters), has been developed for the cell line MDCK. For obtaining a predictive model for the other two cell lines, two future approaches could be followed: (A) introducing the active component of BBB transport into the model or (B) studying more drugs and dividing them into groups depending on the transporter that they bind to, so a different QSPR could be obtained for each transporter acting in the BBB, and its active component would be summarized in each SC.

## 5. Conclusions

A new semi-physiological mathematical model, which includes the three most important factors in brain delivery (barrier transport, the disposition within the brain and drug–brain binding) and is able to predict brain concentration levels in rats for different drugs, has been developed. The best predictions obtained are those that used in vitro data from the MDCK cell line, so these cells together with the new mathematical model could be used as a screening tool by the pharmaceutical industry when a new treatment for a neurological disorder is developed. The model could also be used with data from other more complex cell lines, such as MDCK-MDR1 or hCMEC/D3, although it has been proven that the influence of transporters can bias their predictions. Additional studies are needed, with drug batteries of different transporter substrates, in order to refine these results.

## Figures and Tables

**Figure 1 pharmaceutics-13-01402-f001:**
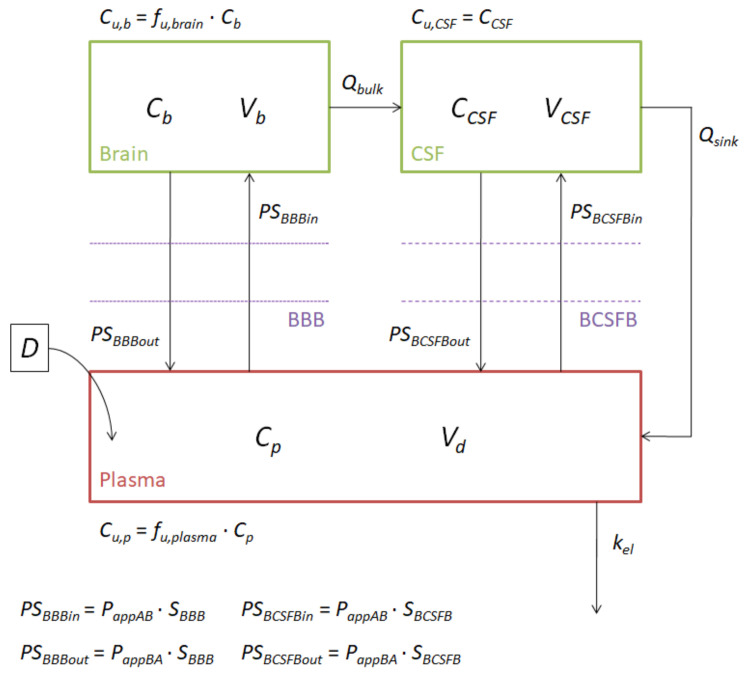
PBPK model scheme.

**Figure 2 pharmaceutics-13-01402-f002:**
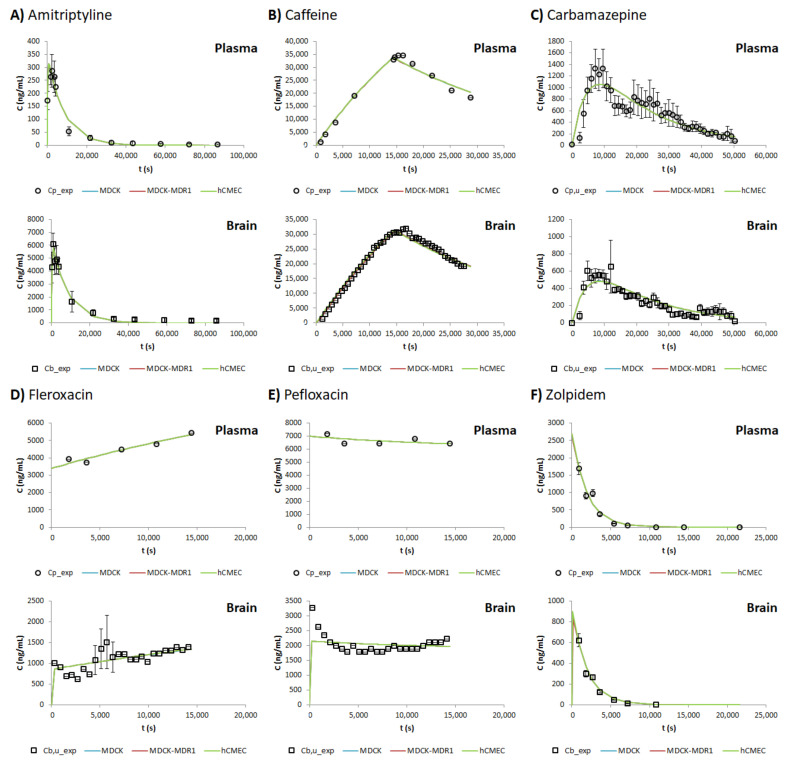
Experimental and fitted plasma and brain profiles for each drug and cell line studied: (**A**) Amitriptyline, (**B**) Caffeine, (**C**) Carbamazepine, (**D**) Fleroxacin, (**E**) Pefloxacin and (**F**) Zolpidem. *Cp_exp* = Experimental total plasma concentration; *Cp,u_exp* = Experimental unbound plasma concentration; *Cb_exp* = Experimental total brain concentration; *Cb,u_exp* = Experimental unbound brain concentration.

**Figure 3 pharmaceutics-13-01402-f003:**
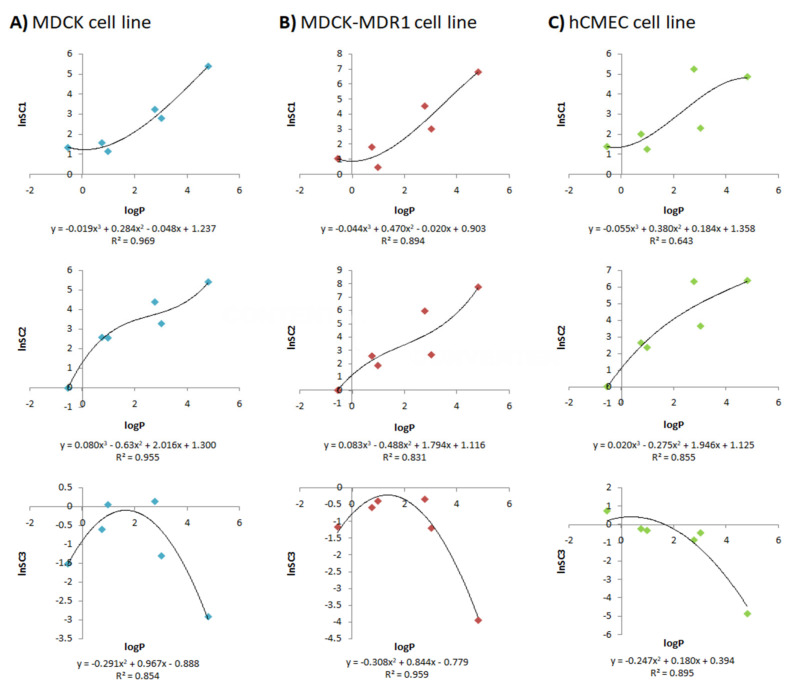
QSPRs developed for each cell line after combining the logP of each drug with the natural logarithm of each scaling factor (LnSC1, LnSC2, LnSC3): (**A**) MDCK cell line, (**B**) MDCK-MDR1 cell line and (**C**) hCMEC/D3 cell line.

**Figure 4 pharmaceutics-13-01402-f004:**
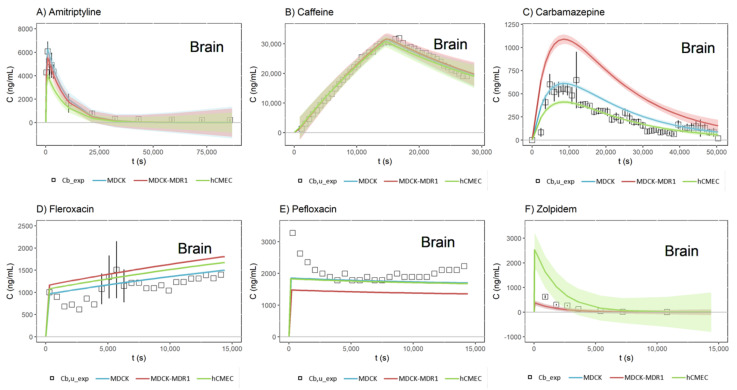
Simulated brain profiles and visual predictive check (VPC, shaded areas above and below fitted lines) for each drug and cell line, after using the QSPR equations: (**A**) Amitriptyline, (**B**) Caffeine, (**C**) Carbamazepine, (**D**) Fleroxacin, (**E**) Pefloxacin and (**F**) Zolpidem. *Cb_exp* = Experimental total brain concentration; *Cb,u_exp* = Experimental unbound brain concentration.

**Table 1 pharmaceutics-13-01402-t001:** PK parameters which were different for each drug.

Drug	*f_u,plasma_*	*V_d_* (cm^3^) *	*k_el_* (s^−1^) *	*k_a_* (s^−1^) *	*D* (ng)	*k_0_* (ng/s)	Rat Weight (g)
Amitriptyline	0.090 ^a^	4000 ^c,+^	7.70 × 10^−5 h^	1.54·10^−4 #^	5,000,000 ^h^		250 ^h^
Caffeine	0.917 ^b^	180 ^c,+^	3.55 × 10^−5 i,$^			833.333 ^i^	300 ^i^
Carbamazepine	0.385 ^b^	1490.5 ^d^	4.50 × 10^−5 j^	8.99·10^−5 #^	3,600,000 ^j^		300 ^j^
Fleroxacin	0.793 ^b^	427.5 ^e,+^	7.13 × 10^−5 k^		1,114,350 ^l^	83.125 ^l^	285 ^l^
Pefloxacin	0.860 ^b^	361.1^f,+^	5.83 × 10^−5 k^		3,676,500 ^l^	214.542 ^l^	285 ^l^
Zolpidem	0.267 ^b^	304 ^g^	1.56 × 10^−4 g^		499,700 ^g^		190 ^g^

* indicates that the parameter was later adjusted; ^+^ indicates that the *V_d_* from the reference was from human data, so it was recalculated for rat; ^$^ this initial estimate corresponds to the α constant rate from the reference; ^#^
*k_a_* initial estimates were calculated as double the *k_el_* initial estimates; ^a^ [23], ^b^ [24], ^c^ [25], ^d^ [26], ^e^ [27], ^f^ [28], ^g^ [20], ^h^ [16], ^i^ [17], ^j^ [18], ^k^ [29], ^l^ [19].

**Table 2 pharmaceutics-13-01402-t002:** Apparent permeability values and the f_u,brain_ obtained from the in vitro tests with MDCK, MDCK-MDR1 and hCMEC cell lines.

Drug	MDCK Cell Line			MDCK-MDR1 Cell Line			hCMEC Cell Line		
	*P_app A→B_* (×10^−6^ cm/s)	*P_app B→A_* (×10^−6^ cm/s)	*f_u,brain_*	*P_app A→B_* (×10^−6^ cm/s)	*P_app B→A_* (×10^−6^ cm/s)	*f_u,brain_*	*P_app A→B_* (×10^−6^ cm/s)	*P_app B→A_* (×10^−6^ cm/s)	*f_u,brain_*
Amitriptyline	74.77 ^a^	178.48 ^a^	0.037 ^a^	17.95 ^a^	16.91 ^a^	0.104 ^a^	124.24 ^b^	66.21 ^b^	0.252 ^b^
Caffeine	26.10	35.31	0.857	33.57	30.59	0.613	63.93	194.70	0.095
Carbamazepine	114.64	78.66	0.673	142.96	75.64	0.238	70.14 ^b^	51.93 ^b^	0.386 ^b^
Fleroxacin	88.48	63.44	0.471	67.40	42.57	0.813	29.96 ^b^	25.73 ^b^	0.743 ^b^
Pefloxacin	41.21	37.49 ^c^	0.910 ^c^	30.82	35.39 ^c^	0.931 ^c^	24.95 ^b^	33.14 ^b^	0.642 ^b^
Zolpidem	21.32	36.48 ^c^	0.971 ^c^	8.92	33.43 ^c^	0.881 ^c^	106.16 ^b^	80.76 ^b^	0.408 ^b^

^a^ Data already published in [11]. ^b^ Data already published in [12]. ^c^ Data already published in [30].

**Table 3 pharmaceutics-13-01402-t003:** PK parameters (*V_d_*, *k_a_*, *k_el_*) and scaling factors (SC1, SC2 and SC3) for each drug and cell line after being adjusted.

Drug	*V_d_* (cm^3^)	*k_el_* (s^−1^)	*k_a_* (s^−1^)	MDCK Cell Line			MDCK-MDR1 Cell Line			hCMEC Cell Line		
				SC1	SC2	SC3	SC1	SC2	SC3	SC1	SC2	SC3
Amitriptyline	14,632.6	1.17 × 10^−4^	5.86 × 10^−3^	220.59	224.82	0.05	920.39	2377.06	0.02	132.89	606.69	0.01
Caffeine	273.6	3.55 × 10^−5^		3.85	1.00	0.22	2.85	1.00	0.31	4.09	1.00	2.15
Carbamazepine	827.0	5.39 × 10^−5^	2.15 × 10^−4^	25.71	81.01	1.16	95.00	391.22	0.71	193.66	569.98	0.44
Fleroxacin	327.3	2.69 × 10^−5^		3.18	12.96	1.07	1.59	6.43	0.68	3.58	10.65	0.74
Pefloxacin	524.3	6.75 × 10^−5^		4.88	13.32	0.55	6.15	13.19	0.56	7.59	14.09	0.81
Zolpidem	185.9	5.12 × 10^−4^		16.53	26.79	0.27	20.59	14.51	0.30	10.26	38.70	0.65

**Table 4 pharmaceutics-13-01402-t004:** The mean brain *C_max_* and AUC PE% for each cell line after fitting the model and after obtaining the simulated profiles using the QSPR equation.

Profile	MDCK Cell Line		MDCK-MDR1 Cell Line		hCMEC Cell Line	
	%PE *C_max_*	%PE AUC	%PE *C_max_*	%PE AUC	%PE *C_max_*	%PE AUC
Fitted	14.34	5.28	15.05	5.45	13.56	5.16
Simulated	19.23	22.34	35.71	48.21	49.77	46.69

%PE = prediction error percentages, *C_max_* = maximum concentration, AUC = area under the curve.

## Data Availability

The authors confirm that the data supporting the findings of this study are available within the article and its Appendix A.

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
