# Peer review of "Physiologically Based Pharmacokinetic (PBPK) Modeling for Predicting Brain Levels of Drug in Rat"

_pharmaceutics, 2021, doi:10.3390/pharmaceutics13091402_

Round 1

Reviewer 1 Report

  • BBB has complicated mechanism whereby it allows compounds through the barrier. Transporters such as Pgp is abundant and therefore it is not only the passive diffusion that governs the permeation at the BBB. The conclusion that results from MDCK cells correlated better than the results from MDCK-MDR1 cells and hCMEC cells is counterintuitive and insufficient explanation is given to explain the phenomenon. MDCK cells will only have passive diffusion and therefore does not represent the BBB appropriately.
  • There is no compound that has increased Papp,B-A in the MDCK-MDR1 cell line. One would expected to see an increased value in these cells since Pgp is expressed. Are all compounds not susceptible to Pgp? In that case, the authors have used a wrong set of model compounds.
  • Fleroxacin concentration-time profile (Figure 2D) in plasma and brain show an upward trend from time 0 till the end of observation period, which is very unusual for a PK profile.

Author Response

  • BBB has complicated mechanism whereby it allows compounds through the barrier. Transporters such as Pgp is abundant and therefore it is not only the passive diffusion that governs the permeation at the BBB. The conclusion that results from MDCK cells correlated better than the results from MDCK-MDR1 cells and hCMEC cells is counterintuitive and insufficient explanation is given to explain the phenomenon. MDCK cells will only have passive diffusion and therefore does not represent the BBB appropriately.

Thanks for your comment. We completely agree with you, but we have used an empiric approximation for obtaining our mathematical model and, in fact, just two drugs susceptible to Pgp were included. In consequence, presently the model represents better the brain access for passive diffusion transported drugs or those P-gp substrates in which the transporter is already saturated.

For obtaining a predictive model for the most complex cell lines, two future approaches could be followed: A) using a mechanistic model, introducing the active component of BBB transport in the model, for which more concentrations should be tested in vitro or B) studying more drugs and dividing them in groups depending on the transporter that they bind to, so a different QSPR could be obtain for each transporter acting in the BBB and its active component would be summarized in each SC.

An explanation of how the empiric model was obtained has been included in the text:

Most of the mathematical predictive models use scaling factors to transform the in vitro data/parameters into in vivo data due to the different characteristics between the in vitro cells/tissues used and the whole organism as for example physiological characteristics as the surface area available for transport. Therefore, every in vitro parameter used in the model needs to have a scaling factor [21,22]. The first and second scaling factors (SC1 and SC2) were used to adjust the in vitro apparent permeabilities to the physiological permeability, as done in [14], because of the differences in terms of surface area and transporters between the in vitro tests and the rat physiology. Moreover, the third scaling factor (SC3) was used to correct the possible deviation present in the in vitro fu,brain values, due to the different composition and behavior of the pig brain homogenate and a healthy rat brain. The use of scaling factors allows researchers to obtain an empiric approximation of how a drug access to the brain. Later on, a refined mechanistic model may be obtained by including both, passive and active transport parameters in the differential equations and the expression levels differences between the in vitro and in vivo situation.

  • There is no compound that has increased Papp,B-A in the MDCK-MDR1 cell line. One would expected to see an increased value in these cells since Pgp is expressed. Are all compounds not susceptible to Pgp? In that case, the authors have used a wrong set of model compounds.

Thanks for the observation. 2 of the 6 compounds used in the model were substrate of Pgp, whose permeability values were previously published. Nonetheless, as just one concentration was tested if the transporter was saturated the increase in Papp,B-A may not be observed.

The information about Pgp susceptibility is included in the supplementary material (table s2), if you consider it relevant we can move it to the main text.

  • Fleroxacin concentration-time profile (Figure 2D) in plasma and brain show an upward trend from time 0 till the end of observation period, which is very unusual for a PK profile.

Profiles were obtained from literature (PMID: 9224772). In that study concentration-time profiles for fleroxacin were obtained after bolus administration (3.91 mg/kg) and infusion (1.05 mg/kg/h), thus the authors aimed at obtaining a steady state brain concentration by keeping the drug infusion rate. If you have any suggestion about any other article with plasma and brain profiles of fleroxacin we will be glad to include those data in our work.

Reviewer 2 Report

 A few areas needs some clarifications.

  • The SC3 (scaling factor 3) used in equation 14 for calculation of brain unbound concentrations Cbu. Recommend author to discuss more about how they determined/ obtained the SC3 for every compound that listed in Tablet 3. Author has given some brief descriptions on the addition of  SC3 in the discussion section ( line 332-337) but detailed information needs to be provided.
  • In Tablet 1 the PK parameters. Several compounds Vd were listed as recalculated from human data to rats ( compounds that are associated with the “+” sign).  Please describe the method for reverse calculation from human to rats. Furthermore, for the same reverse calculation, different rat body weights ( i.e. 250, 285, 300 g) were listed for different compounds that were "reverse calculated".  Please discuss why those/different body weights are used and the impact on the calculations.

Author Response

Thanks for your comments. We are including and change all your suggestions in the manuscript.

  • The SC3 (scaling factor 3) used in equation 14 for calculation of brain unbound concentrations Cbu. Recommend author to discuss more about how they determined/ obtained the SC3 for every compound that listed in Tablet 3. Author has given some brief descriptions on the addition of  SC3 in the discussion section ( line 332-337) but detailed information needs to be provided.

Thanks. More information about the scaling factors has been included in the manuscript.

Most of the mathematical predictive models use scaling factors to transform the in vitro data/parameters into in vivo data due to the different characteristics between the in vitro cells/tissues used and the whole organism as for example physiological characteristics as the surface area available for transport. Therefore, every in vitro parameter used in the model needs to have a scaling factor [21,22]. The first and second scaling factors (SC1 and SC2) were used to adjust the in vitro apparent permeabilities to the physiological permeability, as done in [14], because of the differences in terms of surface area and transporters between the in vitro tests and the rat physiology.  Moreover, the third scaling factor (SC3) was used to correct the possible deviation present in the in vitro fu,brain values, due to the different composition and behavior of the pig brain homogenate and a healthy rat brain. The use of scaling factors allows researchers to obtain an empiric approximation of how a drug access to the brain. Later on, a refined mechanistic model may be obtained by including both, passive and active transport parameters in the differential equations and the expression levels differences between the in vitro and in vivo situation.

The unbound brain concentrations are obtained multiplying the brain concentration by the unbound fraction of drug in brain. The unbound fraction of drug in brain depends on the concentration of proteins in the tissue and the affinity of the drug for those proteins. As different animals may have different amount of proteins in their brains and the homogenization process may damage the proteins of the tissue, the third scaling factor was added for correcting the in vitro fu,brain and making it comparable to the in vivo value.

  • In Tablet 1 the PK parameters. Several compounds Vd were listed as recalculated from human data to rats ( compounds that are associated with the “+” sign).  Please describe the method for reverse calculation from human to rats. Furthermore, for the same reverse calculation, different rat body weights ( i.e. 250, 285, 300 g) were listed for different compounds that were "reverse calculated".  Please discuss why those/different body weights are used and the impact on the calculations.

In those compounds in which the Vd for rats was not found, it was obtained from the human data (L/kg) multiplied by the weight of the rats. As in vivo data from rats were obtained from different articles in literature, we used for the calculation the mean weight that each study had.

A more detailed explanation has been included in the article:

Due to the difficulties for obtaining research articles with all the PK information for rats. In some cases, parameters had to be calculated from human data. The Vd of those drugs marked with a + symbol (table 1) was calculated using the human value (L/kg) and multiplying it by the weight of the rats used in the studies from which the plasma and brain profiles were obtained. Rats weights vary between 250-300 g.

Reviewer 3 Report

The authors present a PBPK model based on in vitro/in vivo data for 4 drugs. The work is interesting and shows the usability of PBPK models to answer difficult or complex "what if" scenarios. However there are some comments. The manuscript is generally difficult to follow regarding the way the results are presented. The introduction, in my opinion, presents more theoretical information that are more or less known to the experienced reader but not so much information what is the actual problem that this work tries to address. Although they may be minor ones I believe that they should be addressed to improve the overall value of this work. That's why I mark it as of "major revisions". 

Comments: 
1) Is it about the PBPK model presentation or evaluation between different cell lines? 
2) line 55. I guess not function but action?
3) Why the authors chooe the specific drugs? 
4) Are the HPLC methods based on previously available validated ones? Please state the respective references. Supplementary. Please provide the partial re-validation parameteres in a table and since HPLC conditions are given, respective chromatograms could be placed too. 
5) Differences of rat brain and cell lines?
6) Figure 2. The experimental data are the literature data and the lines are the model based ones? Not clearly presented in the legend and the lines are all together. Diffiicult to read and follow. 
7) Why we don't see the c-t profiles in the apical-basolateral experiments? How many samples were analyzed for each drug?
8) It is not clearly stated in my opinion if after the QSPR corrections the modes is improved or not. As it stands considering figure 2 and figure 3, all cell lines in figure 2 seem to predict appropriately the in vivo data whereas in figure 3 there are deviations but what does it mean? Are the deviations actually existing or are they differences created from the data processing? Is making it complex better always? See next comment
9)How identical-similar are the cell lines in transporter expression and all the other properties that characterizes them? How the authors ensured similar experimental data? They state that one cell line is better to be applied but are they comparing cell lines with similar characteristics? How sure are they regarding their validation?
10) Please find a way to show and represent not only the fitted values but also the confidence intervals of the model. 

Author Response

Thanks for your comments. We are including and change all your suggestions in the manuscript.

The authors present a PBPK model based on in vitro/in vivo data for 4 drugs. The work is interesting and shows the usability of PBPK models to answer difficult or complex "what if" scenarios. However there are some comments. The manuscript is generally difficult to follow regarding the way the results are presented. The introduction, in my opinion, presents more theoretical information that are more or less known to the experienced reader but not so much information what is the actual problem that this work tries to address. Although they may be minor ones I believe that they should be addressed to improve the overall value of this work. That's why I mark it as of "major revisions". 

Comments: 
1) Is it about the PBPK model presentation or evaluation between different cell lines? 

Thank you for your observation. The objective of the study has been rewritten for clarifying it: Thus, the main objective of this work was to jointly develop a new in vitro system coupled with a physiological-based pharmacokinetic (PBPK) model able to predict brain concentration levels for different drugs in rat. Furthermore, the predictability of the new PBPK model was compared across different cell lines chosen. for obtaining the in vitro neuroPK parameters.

2) line 55. I guess not function but action?

Corrected.

3) Why the authors chooe the specific drugs? 

Due to the difficulties for obtaining research articles with all the PK information for rats, drugs were chosen according to the availability of in vivo data (rat brain concentration-time profiles and plasma concentration-time profiles). Besides that, the inclusion of drugs substrate and non-substrate of Pgp was also considered. We want to give feedback to our model with new in vivo published data.

4) Are the HPLC methods based on previously available validated ones? Please state the respective references. Supplementary. Please provide the partial re-validation parameteres in a table and since HPLC conditions are given, respective chromatograms could be placed too. 

Re-validation parameters and references have been added in supplementary material (table S1).

5) Differences of rat brain and cell lines?

The following paragraph has been included in the manuscript.

For obtaining an ideal in vitro BBB model, an environment reflecting the extracellular matrix and the cells surrounding the endothelial cells that constitute the BBB vessels should be simulated. Nonetheless, this environment has not been yet obtained and cell monolayers lack of pericytes, astrocytes, neurons or different constituents of the neuro-glia vascular unit (NGVU). Because of that, in vitro models can express different transporters present in the BBB but not all. Furthermore, depending on the cell type, in vitro models can express more or less tight junctions avoiding the passage of drugs through the BBB [38]. The main advantage of cell monolayers is its utility for high throughput screening [39].

6) Figure 2. The experimental data are the literature data and the lines are the model based ones? Not clearly presented in the legend and the lines are all together. Diffiicult to read and follow. 

In figure 2, model profiles are overlapping because the scaling factors are fitted to the in vivo data. We have not included a different graph for each cell line, because it would have been too many figures. Nonetheless, if you consider it necessary we can include all the graphs in the supplementary material.

7) Why we don't see the c-t profiles in the apical-basolateral experiments? How many samples were analyzed for each drug?

Those profiles were not included because, as seen in table 2, some of the permeability values were already published and obtained from literature. We have included the permeability value and we consider that we have too many figures.

In those drugs, that are new, three replicates were used in each experiment and four samples were taken (15, 30, 60 and 90 minutes). This information has been included in the manuscript.

8) It is not clearly stated in my opinion if after the QSPR corrections the modes is improved or not. As it stands considering figure 2 and figure 3, all cell lines in figure 2 seem to predict appropriately the in vivo data whereas in figure 3 there are deviations but what does it mean? Are the deviations actually existing or are they differences created from the data processing? Is making it complex better always? See next comment

QSPRs are necessary, because although it may seem that predictions from figure 4 are worse than the predictions from figure 2. In figure 2, the model profiles are fitted to the in vivo data. Nonetheless, as the idea is being able to predict the brain profiles from the in vitro data you need to be able to obtain the scaling factors without fitting them. Thus, we propose the QSPRs.

9)How identical-similar are the cell lines in transporter expression and all the other properties that characterizes them? How the authors ensured similar experimental data? They state that one cell line is better to be applied but are they comparing cell lines with similar characteristics? How sure are they regarding their validation?

Cells are quite different, between them but their use for simulating the BBB was already published and discussed in previous articles of our research group:

In 2013, Mangas Sanjuan et al. proposed a model using two epithelial cell lines, the Madin-Darby canine kidney II (MDCKII) cell line and the wild cell line transfected with P-glycoprotein (MDCKII-MDR1) as, due to their strong tight junctions, they are considered good models for mimicking the BBB. Currently, the endothelial hCMEC/D3 cell line is the best characterized and most used BBB cell model which, despite its relatively lack of tightness (its TEER values are around 30–50 Ω•cm2) is able to overcome some of the main disadvantages of both MDCKII and MDCKII-MDR1 cell lines, as their differences in morphology, growth, metabolism and transporters with human BBB. Although not measured, it is globally accepted that human brain microvessels have TEER values above 1000 Ω•cm2, which would be extremely far from the values detected in hCMEC/D3 monolayers. Nonetheless, previous studies have demonstrated that hCMEC/D3 cells monolayers express several proteins that are responsible of tight junctions’ formation, such as: claudins, occludins or junction adhesion molecules, and they are able to restrict the permeability of lucifer yellow, a low molecular weight paracellular diffusion marker.

The following information has been added to the text:

Table 2 shows the values of the apparent permeabilities (Papp A→B and Papp B→A) and the fu,brain obtained with the MDCK, the MDCK-MDR1 and the hCMEC/D3 cell lines. The Papp A→B corresponds to the Clin of the drug through the BBB, with the combination of Papp A→B and Papp B→A, the Kpuu,brain can be obtained and, using equation 2 the in vitro fu,brain parame-ter can be translated to Vu,brain. So, the three most important neuroPK parameters (Clin, Kpuu,brain and Vu,brain) used to describe the rate and extent of drug delivery to the brain can be obtained with the same methodology, fact that was already pointed in [11] and [12], in-vestigations in which it was proved that the in vivo neuroPK parameters could be pre-dicted from the in vitro ones. In 2013, the methodology was established in MDCK and MDCK-MDR1 cell lines and in 2021 the predictions were compared with the ones obtained in hCMEC/D3 using the same methodology; in all cases, an adequate r2 was obtained [11,12]. Papp A→B, Papp B→A and fu,brain values in table 2 are variable from one cell line to the other which could be due to the morphological and physiological differences between them, both, in terms of tight junctions and transporters [35–37].

10) Please find a way to show and represent not only the fitted values but also the confidence intervals of the model.

Figure 4 has been modified to include the confidence intervals of the model.

Round 2

Reviewer 1 Report

The authors have responded:

"Thanks for your comment. We completely agree with you, but we have used an empiric approximation for obtaining our mathematical model and, in fact, just two drugs susceptible to Pgp were included. In consequence, presently the model represents better the brain access for passive diffusion transported drugs or those P-gp substrates in which the transporter is already saturated."

The above implies that the authors have admitted that their current model does not appropriately describe the normal phenomenon that occurs in the BBB. The authors clearly stated that their model is suited for "passive diffusion transported drugs or ... the transporter is already saturated". Therefore, this model does not benefit the research field of predicting brain drug levels in any way. Prediction of brain penetration is challenging due to the active transporters at the BBB and if the authors have not addressed this in any way and treated the BBB only as a passive diffusion barrier, it is difficult to see how this research will benefit the overall field.

Regarding the previous comment below,

  • There is no compound that has increased Papp,B-A in the MDCK-MDR1 cell line. One would expected to see an increased value in these cells since Pgp is expressed. Are all compounds not susceptible to Pgp? In that case, the authors have used a wrong set of model compounds.

The authors responded that only one concentration was tested and therefore increase in Papp,B-A may not be observed. This response is not sufficient. When a compound is a substrate of Pgp, Papp,B-A would be higher than Papp,A-B regardless of the number of concentration levels tested unless the concentration was so high that saturation of Pgp has occurred. If the tested concentration was indeed saturating Pgp, then the testing condition was flawed and a lower concentration should have been tested. Moreover, the author stated that only 2 of 6 compounds were substrates of Pgp. Again, research in BBB penetration is challenging due to these active transporters and it is difficult to find rationale why the authors chose to work with non-substrate compounds because these would add only little value to advance this field.

Reviewer 3 Report

Thank you for the opportunity to review this work. The authors presented an updated version of their manuscript addressing the comments made from the reviewers. Some comments follow and it is in authors' discretion if they want to address them to further improve their manuscript. 

1) Figures are of very bad resolution from the previous version and I cannot even review them. Please update the resolution and probably it will be better for the quality and the amount of this work that all the graphs would be included in the supplementary file. 
2) Aim of the study to investigate BBB permeability through in vitro-in vivo-in silico approaches. Apologies if I get it wrong but I believe the authors should edit their text and not emphasize which cell line seems to work better for the study of BBB permeability but what cell line feeds better "their" PBPK approach. That's the reason why there were comments about in vitro/in vivo differences, cell-lines differences etc. in my initial report, as well as relative comments from the other reviewers. I believe their work will be better if presented as a developed PBPK model that requires in vitro data from MDCK than other lines. 
3) The PBPK model was developed in what software? R? Matlab? Please state it.